# Strategies for retention of heterosexual men in HIV care in sub-Saharan Africa: A systematic review

Sylvia Kusemererwa[1]*, Dickens Akena[2], Damalie Nakanjako[3], Joanita Kigozi[4], Regina Nanyunja[1], Mastula Nanfuka[5], Bennet Kizito[6], Joseph Mugisha Okello[1], Nelson Kawulukusi Sewankambo[3]

1 Department of HIV Interventions, Medical Research Council/Uganda Virus Research Institute (MRC/UVRI) and London School of Hygiene and Tropical Medicine (LSHTM), Uganda Research Unit, Entebbe, Uganda, 2 Department of Psychiatry, School of Medicine, Makerere University College of Health Sciences, Kampala, Uganda, 3 Department of Medicine, School of Medicine, Makerere University College of Health Sciences, Kampala, Uganda, 4 Department for Outreaches, Infectious Diseases Institute (IDI), Makerere University College of Health Sciences, Kampala, Uganda, 5 Department of TBSpeed, Makerere University Johns Hopkins University Research Collaboration (MUJHU), Kampala, Uganda, 6 Department of Monitoring and Evaluation, The AIDS Support Organization (TASO), Kampala, Uganda

* Sylvia.Kusemererwa@mrcuganda.org, kushylvia@gmail.com

**Data Availability Statement:** All relevant data are within the manuscript and its Supporting information files.

## Abstract

Expansion of Antiretroviral Therapy (ART) programs in sub-Saharan Africa (SSA) has increased the number of people accessing treatment. However, the number of males accessing and being retained along the human immunodeficiency virus (HIV) care cascade is significantly below the UNAIDS target. Male gender has been associated with poor retention in HIV care programs, and little is known about strategies that reduce attrition of men in ART programs. This review aimed to summarize any studies on strategies to improve retention of heterosexual males in HIV care in SSA. An electronic search was conducted through Ovid® for three databases (MEDLINE®, Embase and Global Health). Studies reporting interventions aimed at improving retention among heterosexual men along the HIV care cascade were reviewed. The inclusion criteria included randomized-controlled trials (RCTs), prospective or retrospective cohort studies that studied adult males (≥15years of age), conducted in SSA and published between January 2005 and April 2019 with an update from 2019 to 2020. The search returned 1958 articles, and 14 studies from eight countries met the inclusion criteria were presented using the PRISMA guidelines. A narrative synthesis was conducted. Six studies explored community-based adherence support groups while three compared use of facility versus community-based delivery models. Three studies measured the effect of national identity cards, disclosure of HIV status, six-monthly clinic visits and distance from the health center. Four studies measured risk of attrition from care using hazard ratios ranging from 1.2–1.8, four studies documented attrition proportions at an average of 40.0% and two studies an average rate of attrition of 43.4/1000PYs. Most (62%) included studies were retrospective cohorts, subject to risk of allocation and outcome assessment bias. A pooled analysis was not performed because of heterogeneity of studies and outcome definitions. No studies have explored heterosexual male- centered interventions in HIV care. However, in included studies that explored retention in both males and

**Funding:** This review was funded by Afya Bora Consortium, supported by PEPFAR & HRSA, Grant number U91HA06801

**Competing interests:** The authors have declared that no competing interests exist.

females, there were high rates of attrition in males. More male-centered interventions need to be studied preferably in RCTs. Registry number: PROSPERO2020 CRD42020142923 Available from: https://www.crd.york.ac.uk/prospero/display_record.php?ID=CRD42020142923.

## Introduction

According to the UNAIDS report 2018, mortality due to HIV/AIDS among men is higher compared to women [1]. In 2017, an estimated 300 000 [220 000–410 000] men in sub-Saharan Africa (SSA) died of acquired immune deficiency syndrome (AIDS)-related illness compared to 270 000 [190 000–390 000] women. Although women bear the highest burden of disease in SSA, more men than women living with Human immunodeficiency virus (HIV) are dying [2]. Low treatment coverage among men and poor treatment seeking behavior have been sighted as some of the reasons for the higher mortality of men with HIV compared to women [3].

Expansion of Antiretroviral Therapy (ART) programs in SSA has greatly increased the number of people accessing treatment [4]. In Uganda for example, 1.4million people were living with HIV in 2018 and 73% were on ART [5]. For adequate viral suppression to be realized, patients need to adhere to ART over their lifetime [6]. Retention in care is key to achieving the milestones that have been set up in HIV care [7]. According to the World Health Organization (WHO), retention in care can be defined from the moment of initial engagement in care, when a person with HIV is linked successfully to services, to assessment for eligibility, initiation on ART and retention in lifelong ART care [8]. However, data from high-income countries (HIC) as well as low and middle-income countries (LMIC), SSA inclusive, have shown a significant reduction in patient retention in HIV care at each step of the HIV care continuum. The continuum starts from diagnosis and linkage to care, assessment of ART readiness to acceptability, receipt of ART, adherence and retention in care, and treatment success as indicated by virologic suppression [3,9]. Attrition has been particularly documented in younger men, especially those less than 35 years) [10]. The high rates of attrition in males continue to peg down the gains made in HIV care over the years.

Several studies have been done to assess strategies that improve retention of those in HIV care [11,12]. Studies that include HIV positive men have shown improved rates of retention (>80%) using community-based strategies and reduction of clinic contact visits [13,14]. However, other studies showed a high risk of attrition among males compared to females [adjusted hazard ratio (aHR) range from 1.2–1.8] [15,16]. Interestingly, research exploring the use of mobile text to support retention documented no differences between males and females in using mobile text messages to support retention [13,17–19]. In one study, conducted in Kenya and Uganda, males were found to require more tracing to support retention [13]. Attrition along the HIV cascade could slow down the gains in mitigating the HIV epidemic in SSA [9].

Generally, the number of males accessing and being retained along the HIV care cascade is lower compared to women [20]. Male gender has been associated with poor retention in care [21]. Furthermore, there is conflicting literature about different strategies to improve retention of men in care. We conducted a systematic review to summarize any studies on strategies to improve retention of heterosexual males (≥15years of age) in HIV care in SSA. This review aimed at informing policy, research and practice on retention of HIV positive males in HIV care in SSA.

The aim of the systematic review was to identify, synthesize and appraise existing evidence of interventions aiming to improve retention of heterosexual men in HIV care in sub-Saharan Africa. The research question broken down by PICOS criteria was [22]:

P—population—men living with HIV in SSA

I—intervention—interventions that aimed to improve retention among men living with HIV in SSA

O—outcome—Studies that documented retention proportion/rate, attrition rate/proportion, relative risk, hazard ratios, odds ratios or retention strategies

S—study design—Randomized controlled trials, controlled clinical trials (CCT), prospective cohort studies, and retrospective cohort studies

## Materials and methods

This review was conducted and reported according to the Preferred Reporting Items for Systematic Reviews and Meta-analysis (PRISMA) statement [23]. A search of studies published between January 2005 and April 2019 was conducted and updated to include articles published from May 2019 to December 2020. This period was selected because ART scale up in SSA started around 2005. Data on strategies that improve retention of men in HIV care in any country in SSA was extracted. The search outputs were summarised in a PRISMA flow chart (Fig 1).

### Inclusion and exclusion criteria

Studies were included if they met the following criteria: published or presented between January 2005 and April 2019 and conducted in SSA. We included randomized controlled trials and cohort studies that recruited adult males ($\geq$15years of age). Studies were excluded if they had other study designs (qualitative, cross-sectional, case control). We also excluded previous systematic reviews, articles that measured other outcomes other than retention, conference abstracts and letters to the editor.

### Review protocol

The review protocol was registered in PROSPERO, the International Prospective Register of Systematic Reviews on 29 April 2020. The registration number is CRD42020142923 and can be found online: https://www.crd.york.ac.uk/prospero/display_record.php?ID=CRD42020142923.

### Data sources and search strategies

Studies were identified through a systematic search in bibliographic databases; using the OVID® interface for (MEDLINE, EMBASE and Global Health). Search terms were developed using relevant key words and medical subject headings (MeSH). The search terms are summarized in S1 Table. Duplicate studies were removed from the results using the "de-duplicate" feature in OVID® and exported to EndNote reference management software (X7).

### Screening and selection

Two reviewers (SK and RN) reviewed the identified studies (titles, abstracts, keywords) independently for the eligibility criteria and articles that did not meet the eligibility criteria were

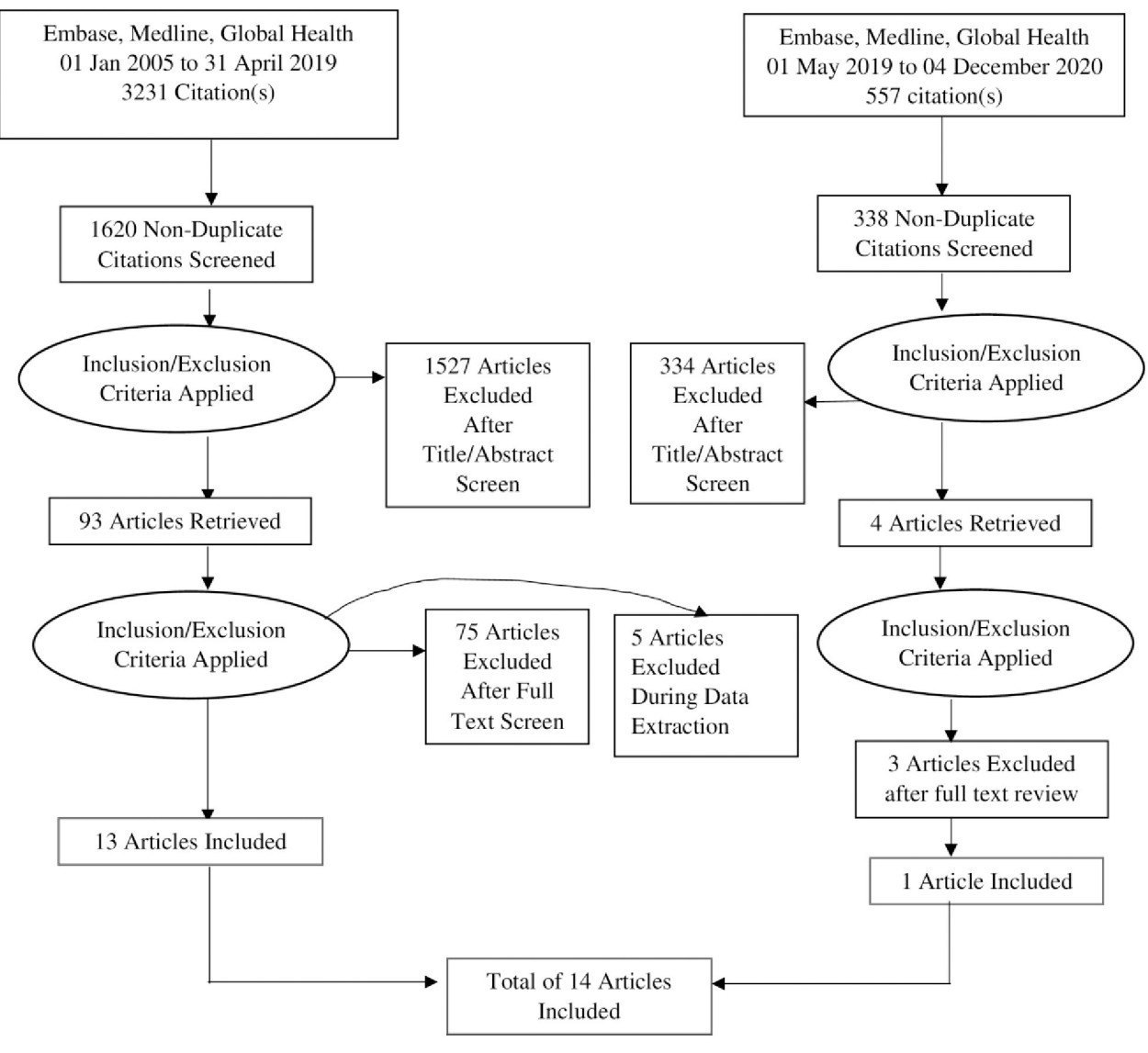

**Fig 1. PRISMA flow chart for study selection.**

excluded at this stage. Articles from stage one were subjected to full text review in which the methods were reviewed for relevance towards the eligibility criteria. More articles were excluded at this stage. Two copies of full text documents were printed for all articles that met eligibility criteria. SK and RN reviewed these independently and were blinded to the results from each other. Any inconsistencies between were discussed and where consensus was not reached, a discussion was held with a third member of the team. Any missing information was not collected from any corresponding authors.

## Data extraction

A standardized data extraction form was used to document information from each included study on the following: Data items collected included (a) first author's name, (b) publication year, (c) country, (d) sample size for males and total population, (e) study design and (f) interventions used to promote retention. Information on outcome measures was also collected.

### Risk of bias assessment

Risk of bias within the included studies was assessed using the Cochrane risk of bias tool for non-randomized studies in interventions (ROBINS-I) [24].

### Summary measures

We sought to establish retention proportion/rate, attrition rate/proportion, relative risk, hazard ratios, and odds ratios.

### Data synthesis

Due to the heterogeneity of studies a meta-analysis was not considered and narrative synthesis of the included papers was conducted. Results were summarized using PRISMA flowchart [23].

### Ethics statement

This work did not require an ethics statement.

## Results and discussion

### Summary of search results

A total of 1620 records were identified through the literature search. Titles and abstracts were reviewed and 1570 articles were excluded because they did not meet the inclusion criteria. An updated search identified 557 records of which 334 were excluded because they did not meet the inclusion criteria.

### Study selection

A total of 93 articles were subjected to full text review to determine eligibility (Fig 1), 18 met eligibility. During data extraction, five articles were excluded because they were protocols for planned studies. Therefore, 13 studies were included in this review. An updated search was conducted on 04 December 2020 and 4 additional records were subjected to full text review, and 3 articles were excluded because two did not have clear description of interventions and one did not disaggregate the outcomes by gender. Only one article was include bringing the total number of articles included in this review to 14.

### Study characteristics

A summary description of the included studies is presented in Table 1 (adapted from the Cochrane library) [25]. The studies were conducted in eight African countries: Democratic Republic of Congo, Ethiopia, Malawi, Mozambique, Rwanda, South Africa, Tanzania and Uganda between 2005 and December 2020. The number of males in included studies varied and ranged from 122 to 21101, a proportion of 32% of the total population.

### Risk of bias within studies

The quality of each study was assessed using the Cochrane risk of bias tool for non-randomized studies [24]. The risk of bias with regards to the different domains within included studies varied from low, moderate to serious. With regards to confounding, there was low to moderate risk of bias (Table 2). The bias in methods for selection of study participants was moderate in 10 studies [15,16,28,29,31–36], low in 3 [27,30,37], while there was a serious risk of bias due to absence of data on some variables that were considered confounders [26]. There was bias in

**Table 1. Studies that met inclusion criteria.**

| Author, year, Country | Journal | Sample size Males (Total) | Age (years) | Study design | Intervention | Outcome measure | RetentionMales (Female) |
|---|---|---|---|---|---|---|---|
| Decroo, 2017, Mozambique [15] | BMJ Open | 884(2406) | ≥15 | Retrospective Cohort | Joining Community ART Groups (CAGs) | Retention at 12 and 24 months | 88.2(92.4), 80.8 (88.9) Risk of attrition (aHR: 1.80, 95%CI 1.41–2.51) |
| Wringe, 2018, Malawi [16] | JIAS | 7695 (22633) | ≥18 | Retrospective Cohort analysis | Six-monthly clinical consultation schedule | Attrition rate | 37.9/1000pys (30.5/ 1000pys) aHR = 1.3 |
| Decroo, 2014, Mozambique [26] | Tropical Medicine and International Health | 1746(5729) | 30–43 | Retrospective Cohort | Joining Community ART Groups | Attrition proportion, HRs | Attrition = 47.8 uHR = 2.07(95%CI: 1.59–2.70) |
| | | | | | | | aHR = 1.93(95%CI: 1.48–2.51) |
| Fatti, 2012, South Africa [27] | Implementation and Operational Research: Clinical Science | 21101 (66953) | 29.4–42.3 | Observational Cohort | Receiving community based adherence support | LTFU | uHR = 1.23 (95%CI: 1.16–1.30) |
| | | | | | | | aHR = 1.34 (95%CI: 1.24–1.44) |
| Nabaggala, 2018, Uganda [28] | BMC Research Notes | 122(381) | 23-35 (IQR) | Retrospective Cohort | Tracking of PLHIV | Return to clinic proportion | 56.6 (76.9) |
| Rich, 2012, Rwanda [29] | Implementation and Operational Research: Clinical Science | 349(1041) | ≥18 | Retrospective Cohort | Enrolment in a community based ART program | Retention proportion | 32.1(67.8) |
| | | | | | | Attrition proportion | 40 (40) |
| Tsondai, 2017, South Africa [30] | JIAS | 948(3216) | ≥16 | Retrospective Observational Cohort | Enrolment in adherence clubs | LTFU HRs | No difference |
| Kipp, 2012, Uganda [31] | PLOS One | 163(385) | ≥18 | Comparative Cohort | Community Vs Facility based ART delivery program | LTFU proportions | Community based = 50.0% (28/56) |
| | | | | | | | Facility based = 48.3% (28/58). p = 0.854 |
| Megereso, 2016, Ethiopia [32] | BMC Health Services Research | 834(1895) | ≥18 | Retrospective Cohort | Treatment in a primary health center Vs Hospital | Survival HR | aHR = 1.4 (95%CI: 1.1–1.7) |
| Akilimali, 2017, DRC [33] | PLOS One | 238(717) | >18 | Cohort | Disclosure of HIV status | LTFU rate per 1000pys | 48.9 (25.5) |
| Siril, 2017, Tanzania [34] | AIDS Research and Therapy | 208(824) | ≥18 | Prospective Cohort | NAMWEZA"Yes, together we can" Receiving psychosocial support | LTFU | Male gender was associated with higher risk of LTFU, p = 0.04† |
| Shearer, 2016, South Africa [35] | BMJ Open | 4943 (12219) | ≥18 | Observational Cohort | Reporting identification status | Attrition proportion | 23.0 (15.8) |
| Bilinski, 2017, Malawi [36] | PLOS One | 1422(3949) | 33 (mean) | Retrospective Cohort | Travel distance to health center for care | Hazard ratio (HR) | uHR = 1.64 (95%CI: 1.46 ±1.84) |
| | | | | | | | aHR = 1.62 (95% CI: 1.44 ±1.82) p<0.0001 |
| Bock, 2019, South Africa [37] | JIAS | 166 (465) | ≥18 | Retrospective Cohort Analysis | Referral to adherence Clubs | LTFU (HR) | No difference by genderaHR 1.09 (95% CI:0.7–1.69) p = 0.704 |

aHR = adjusted hazard ratio, BMC = Biomedical central, BMJ = British Medical Journal, CI = Confidence interval, JIAS = Journal of International AIDS Society,

IQR = Inter quartile range, LTFU = Loss to follow up, PLHIV = person living with HIV, PY = Person years, uHR- Unadjusted hazard ratio.

† Measure of association not provided for males.

some studies due to misclassification of interventions due to recall bias from study participants in one study and [33] and missing data in others [15,16,27–32,34–37].

The risk of bias due to deviations from interventions was generally moderate with two studies having low risk of bias [33,37] and another providing no information on deviations

**Table 2. Risk of bias within studies.**

| | | Study | | | | | | | | | | | | | | |
|---|---|---|---|---|---|---|---|---|---|---|---|---|---|---|---|---|
| | | Decroo, 2017[15] | Wringe, 2018[16] | Dercoo, 2014[26] | Fatti, 2012[27] | Nabaggala, 2018[28] | Rich, 2012[29] | Tsondai, 2017[30] | Kipp, 2012[31] | Megereso, 2016[32] | Akilimali, 2017[33] | Siril, 2017[34] | Shearer, 2016[35] | Bilinski, 2017[36] | Bock, 2019[37] |
| **Risk of bias domains** | Confounding | - | - | - | + | - | - | - | - | - | - | - | - | - | - |
| | Bias in selection of participants into the study | - | - | x | + | - | - | + | - | - | - | - | - | - | + |
| | Bias classification of intervention | + | - | - | - | - | - | - | - | - | - | - | - | x | + |
| | Bias due to deviations from intended interventions | - | - | - | - | - | - | - | - | - | + | - | - | - | + |
| | Bias due to missing data | - | - | - | - | - | - | - | x | - | x | - | + | - | - |
| | Bias in measurement of outcomes | - | - | - | - | - | - | - | - | - | - | - | + | - | - |
| | Bias in selection of the reported result | + | + | + | + | + | + | - | + | + | - | + | + | + | + |

Key.

+ = Low risk of bias: Low risk of bias - the study is comparable to a well-performed randomized trial with regard to this domain.

– = Moderate risk of bias: Moderate risk of bias–the study is sound for a non-randomized study with regard to this domain but cannot be considered comparable to a well-performed randomized trial.

X = Serious risk of bias: Serious risk of bias–the study has some important problems in this domain.

Critical risk of bias–the study is too problematic in this domain to provide any useful evidence on the effects of intervention.

No information on which to base a judgement about risk of bias for this domain.

[15,16,26–32,34–36]. There was moderate bias due to missing data but most studies adjusted for this in the analysis. However, one study had serious risk of bias in the same domain because they relied on patients to provide information on disclosure [33] while another indicated the lack of information as a limitation for the study [28].

The bias in measurement of outcomes was moderate in most studies as most authors adjusted for the factors that may affect the outcome in data analysis. The bias in selection of the reported result was moderate to low. In most studies, the results were reported as indicated while two studies had moderate risk of bias in this domain as there was no comparison of outcomes in the two groups being studied [30,33].

## Retention strategies and outcome measures

In this review, seven studies (7/14) explored community based adherence support groups [15,26–30,37] while two (2/14) compared use of facility versus community-based delivery models [31,32]. Other studies measured the effect of disclosure of HIV status [33], giving psychosocial support [34], national identity cards [35], distance from the health center [36] and six-monthly clinic visits [16]. Retention of men was provided as proportions in two studies [26,29], an average of 56.5% at 24months. Five studies provided risk of attrition from care using hazard ratios ranging from 1.2–1.8 [15,26,27,32,36], four studies documented attrition proportions at an average of 40.0% [26,29,31,35] and two studies an average rate of attrition of 43.4/1000 person years [16,33].

The engagement of men in HIV care is important in ensuring epidemic control and achievement of UNAIDS targets [20]. Therefore, it is necessary to identify and set up strategies that will effectively ensure the retention of men in HIV care [38]. This review showed that few studies have investigated interventions that can be used to retain heterosexual males in HIV care. What is interesting is no study was found that focused on men as a population. Various studies have focused on involving men in PMTCT programs [39,40].

We also found that in this review the highest proportion of men retained in care was observed when community adherence groups were used as an intervention. Although retention in the population was high, the risk of attrition of men from care was at 80% [15]. Other studies also showed a high risk of attrition for men [16,26,27,32,36]. This is similar to what has been studied as predictors for poor retention in various studies in sub Saharan Africa where male gender has been highlighted as a risk factor for mortality and attrition from care [41–44]. Whereas retention is a bigger challenge for men relative to women, various studies did not provide a disaggregation of retention by gender; posing a challenge in identifying interventions that may work for men [45–50]. In contrast, most studies on engaging men who have sex with men have been undertaken outside SSA [51–55] with retention reported at about 64% [50]. A study done in Kenya showed that men who have sex with men (MSM) had lower retention at 12 months compared to heterosexual men and women when ART was received at the clinic compared to when it was not [56].

Although retention in care has been posed as a general challenge in SSA, most especially among men [44] similar sentiments have been seen elsewhere in the developed world like the United States of America where disparities in retention have been reported [57]. Gender and race account for differences in retention and also affect access to care with males and blacks being at higher risk of discontinuing care [58]. However, studies among those disproportionately affected by HIV have shown some improvement in retention in care with community based interventions facilitating engagement in care [59] while mobile phone used improved retention of these groups from 51% to 81% at 12 months [60].

### Strengths and limitations

To the best of our knowledge, this is the first systematic review aimed at summarizing strategies to reduce attrition of men from HIV care. However, there are a few limitations to this review. Firstly, only a few studies on retention of heterosexual men in HIV care have been performed and only eight countries in sub Saharan Africa are represented. Eight of these were in southern Africa and six in East Africa. This may limit the generalizability of the findings to central and western Africa that may have different gender-related sociocultural practices. Additionally, the outcome measures used in the identified studies varied widely, making it impractical to perform a pooled analysis.

It is important to note that all the studies that included interventions to improve retention of men in HIV care were retrospective cohorts, making it challenging to account for the effects of confounding factors. While randomized controlled trials (RCT) are the gold standard in evaluating interventions including their effect sizes [61], none of the included studies in this review was an RCT. Therefore, the evidence from this review should be interpreted with caution.

Lastly, we didn't explore whether the males lost in one program were identified or seen in another. Most of the studies used programmatic data that may not provide the true estimates for retention of men in care.

## Conclusions

This review suggests that no studies have explored heterosexual male centered interventions in HIV care. However, in included studies that explored retention in both males and females, there were high rates of attrition in men. The barriers and facilitators for retention of men in HIV care need to be explored in order to design male-centered interventions in SSA. There is also need to study the effectiveness of potentially effective strategies, preferably through randomized controlled trials and any interventions put in place should be evaluated.

## Supporting information

**S1 Checklist.**
(DOC)

**S1 Table. Search term strategy used in Ovid® for three databases (Medline, Embase and Global Health).**
(DOCX)

## Acknowledgments

The authors wish to thank the Afya Bora Consortium working group for all the support and guidance offered during the conduct of this review. They would also want to thank Russell Burke of London School of Hygiene and Tropical Medicine for providing enormous support and guidance on library searches.

## Author Contributions

**Conceptualization:** Sylvia Kusemererwa, Dickens Akena, Damalie Nakanjako, Joanita Kigozi, Regina Nanyunja, Mastula Nanfuka, Bennet Kizito, Joseph Mugisha Okello, Nelson Kawulukusi Sewankambo.

**Data curation:** Sylvia Kusemererwa, Dickens Akena, Regina Nanyunja, Bennet Kizito, Joseph Mugisha Okello.

**Formal analysis:** Sylvia Kusemererwa, Dickens Akena, Damalie Nakanjako, Regina Nanyunja, Joseph Mugisha Okello.

**Investigation:** Sylvia Kusemererwa, Dickens Akena, Regina Nanyunja, Joseph Mugisha Okello, Nelson Kawulukusi Sewankambo.

**Methodology:** Sylvia Kusemererwa, Dickens Akena, Damalie Nakanjako, Regina Nanyunja, Mastula Nanfuka, Bennet Kizito, Joseph Mugisha Okello, Nelson Kawulukusi Sewankambo.

**Project administration:** Sylvia Kusemererwa.

**Resources:** Sylvia Kusemererwa.

**Supervision:** Sylvia Kusemererwa, Dickens Akena, Damalie Nakanjako, Joanita Kigozi, Joseph Mugisha Okello, Nelson Kawulukusi Sewankambo.

**Validation:** Sylvia Kusemererwa, Dickens Akena, Regina Nanyunja, Joseph Mugisha Okello.

**Visualization:** Sylvia Kusemererwa, Dickens Akena, Regina Nanyunja, Joseph Mugisha Okello.

**Writing – original draft:** Sylvia Kusemererwa, Dickens Akena, Damalie Nakanjako, Joanita Kigozi, Regina Nanyunja, Mastula Nanfuka, Bennet Kizito, Joseph Mugisha Okello, Nelson Kawulukusi Sewankambo.

**Writing – review & editing:** Sylvia Kusemererwa, Dickens Akena, Damalie Nakanjako, Joanita Kigozi, Regina Nanyunja, Mastula Nanfuka, Bennet Kizito, Joseph Mugisha Okello, Nelson Kawulukusi Sewankambo.

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
