## [Decision Letter · Decision Letter 0]

1 Dec 2020

PONE-D-20-19143

Strategies for retention of heterosexual men in HIV care in sub-Saharan Africa: A systematic review

PLOS ONE

Dear Dr. Sylvia Kusemererwa,

Thank you for submitting your manuscript to PLOS ONE. After careful consideration, we feel that it has merit but does not fully meet PLOS ONE’s publication criteria as it currently stands. Therefore, we invite you to submit a revised version of the manuscript that addresses the points raised during the review process.

We look forward to receiving your revised manuscript.

Kind regards,

Claudia Marotta

Academic Editor

PLOS ONE

Journal Requirements:

2. We note that your literature search was performed on April 2019;to allow an up-to-date view of the topic, we would request that the search is updated.

3.In your Data Availability statement, you have not specified where the minimal data set underlying the results described in your manuscript can be found. PLOS defines a study's minimal data set as the underlying data used to reach the conclusions drawn in the manuscript and any additional data required to replicate the reported study findings in their entirety. All PLOS journals require that the minimal data set be made fully available. For more information about our data policy, please see http://journals.plos.org/plosone/s/data-availability.

4. We noted in your submission details that a portion of your manuscript may have been presented or published elsewhere.

[No, except for the protocol that was published in PROSPERO]

Please clarify whether this conference proceeding or publication was peer-reviewed and formally published. If this work was previously peer-reviewed and published, in the cover letter please provide the reason that this work does not constitute dual publication and should be included in the current manuscript.

Additional Editor Comments (if provided):

Dear Authors follow reviewers suggestion to improve your article

Reviewers' comments:

Reviewer's Responses to Questions

**Comments to the Author**

1. Is the manuscript technically sound, and do the data support the conclusions?

Reviewer #1: Yes

Reviewer #2: Yes

2. Has the statistical analysis been performed appropriately and rigorously? 

Reviewer #1: Yes

Reviewer #2: N/A

3. Have the authors made all data underlying the findings in their manuscript fully available?

Reviewer #1: Yes

Reviewer #2: Yes

4. Is the manuscript presented in an intelligible fashion and written in standard English?

Reviewer #1: Yes

Reviewer #2: Yes

5. Review Comments to the Author

Reviewer #1: The authors have used meticulous methods to identify data on interventions to keep men engaged in HIV care and treatment programs. This topic is of great public health importance in order to control the HIV epidemic. Unfortunately there is lack of RCT evidence on interventions. None the less authors did a thorough job and should be commended.

Reviewer #2: The authors present a well written and rigorously conducted systematic review on the important topic of retention of heterosexual men in SSA. I have only a few minor comments:

Line 140 notes that articles among the excluded were "articles that did not measure the outcomes other than retention". I think this is confusing and the authors intend to exclude articles that measured outcomes other than retention.

Also the text shows the range of population of males in the examined studies. A description of the percentages of males would be of value in this section (around line 194).

If males are more mobile, it may be that attrition or lost to follow up at a particular clinic or program may be followed by a resumption of care elsewhere and true retention deficits are overestimated. This is a challenge of much programmatic data because it is hard to understand true rates of attrition, is probably a shortcoming of included articles, and is probably worth addressing here.

It may be interesting to note how SSA continuum outcomes compare to those in the US - that retention is not just a problem in SSA and some of the lessons on retention could be evaluated (with cultural caveats) for strategies to address gaps in the care cascade in resource rich settings as well. This is just a suggestion for consideration.

6. PLOS authors have the option to publish the peer review history of their article (what does this mean?). If published, this will include your full peer review and any attached files.

Reviewer #1: **Yes: **Moses H Bateganya

Reviewer #2: No

---

## [Author Response · Author response to Decision Letter 0]

19 Dec 2020

RESPONSE TO REVIEWERS FOR PONE-D-20-19143

Strategies for retention of heterosexual men in HIV care in sub-Saharan Africa: A systematic review

PLOS ONE

ADDITIONAL REQUIREMENTS: 

Comment 1: Please ensure that your manuscript meets PLOS ONE's style requirements, including those for file naming. The PLOS ONE style templates can be found at

https://clicktime.symantec.com/3N5r6sHwWDQ4EZGHWqnSbX46H2?u=https%3A%2F%2Fjournals.plos.org%2Fplosone%2Fs%2Ffile%3Fid%3DwjVg%2FPLOSOne_formatting_sample_main_body.pdf and

https://clicktime.symantec.com/3VVZjeLDp4JvUgfMCbYnWnW6H2?u=https%3A%2F%2Fjournals.plos.org%2Fplosone%2Fs%2Ffile%3Fid%3Dba62%2FPLOSOne_formatting_sample_title_authors_affiliations.pdf

Response: The manuscript has been updated to conform to the PLOS ONE style requirements.

Comment 2: We note that your literature search was performed on April 2019; to allow an up-to-date view of the topic, we would request that the search is updated.

Response: We thank the editor for this suggestion. The search was updated to include articles from May 2019 to 04 December 2020. One study that met the inclusion criteria has been added to the number of selected articles.

“Bock P, Gunst C, Maschilla L, Holtman R, Grobbelaar N, Wademan D, Dunbar R, Fatti G, Kruger J, Ford N, Hoddinott G. Retention in care and factors critical for effectively implementing antiretroviral adherence clubs in a rural district in South Africa. Journal of the International AIDS Society. 2019 Oct;22(10):e25396.”

Comment 3: In your Data Availability statement, you have not specified where the minimal data set underlying the results described in your manuscript can be found. PLOS defines a study's minimal data set as the underlying data used to reach the conclusions drawn in the manuscript and any additional data required to replicate the reported study findings in their entirety. All PLOS journals require that the minimal data set be made fully available. For more information about our data policy, please see https://clicktime.symantec.com/39fpXW1RseZwd5kGYGwRABY6H2?u=http%3A%2F%2Fjournals.plos.org%2Fplosone%2Fs%2Fdata-availability.

Response: We wish to clarify that this being a systematic review, the minimal data set for this review is summarized in the PRISMA Chart on page 7, data extraction table on page 11-12 and Supporting Information table on page 24-25, which are part of the manuscript.

Upon re-submitting your revised manuscript, please upload your study’s minimal underlying data set as either Supporting Information files or to a stable, public repository and include the relevant URLs, DOIs, or accession numbers within your revised cover letter. For a list of acceptable repositories, please see https://clicktime.symantec.com/3TF1hioN2rzn1pqeLNFGmcx6H2?u=http%3A%2F%2Fjournals.plos.org%2Fplosone%2Fs%2Fdata-availability%23loc-recommended-repositories. Any potentially identifying patient information must be fully anonymized.

Response: We wish to clarify that this being a systematic review, the minimal data set for this review is summarized in the PRISMA Chart on page 7, data extraction table on page 11-12 and Supporting Information table on page 24-25, which are part of the manuscript.

Important: If there are ethical or legal restrictions to sharing your data publicly, please explain these restrictions in detail. Please see our guidelines for more information on what we consider unacceptable restrictions to publicly sharing data: https://clicktime.symantec.com/38ewnGthVPixJ2HJiBabjjh6H2?u=http%3A%2F%2Fjournals.plos.org%2Fplosone%2Fs%2Fdata-availability%23loc-unacceptable-data-access-restrictions. Note that it is not acceptable for the authors to be the sole named individuals responsible for ensuring data access.

Response: We wish to clarify that there are no ethical or legal restrictions to sharing our data publicly as all of it is from already published data.

 Response: We appreciate the editor for this comments and suggestion.

Comment 4: We noted in your submission details that a portion of your manuscript may have been presented or published elsewhere.

[No, except for the protocol that was published in PROSPERO]

Please clarify whether this conference proceeding or publication was peer-reviewed and formally published. If this work was previously peer-reviewed and published, in the cover letter please provide the reason that this work does not constitute dual publication and should be included in the current manuscript.

Response: We appreciate the editor for this observation. As a requirement for a systematic review, only the protocol was registered and formally published at PROSPERO, the registration number is: CRD42020142923, Available from:

https://www.crd.york.ac.uk/prospero/display_record.php?ID=CRD42020142923. PROSPERO does not necessary peer review protocols, but checks if they conform to their format/guidelines. We add that this work does not constitute dual publication. This information has been added in the cover letter.

COMMENTS FROM REVIEWERS AND RESPONSES

We appreciate the reviewers for their comments provided. Below are responses to their suggestions

Reviewer #1: The authors have used meticulous methods to identify data on interventions to keep men engaged in HIV care and treatment programs. This topic is of great public health importance in order to control the HIV epidemic. Unfortunately, there is lack of RCT evidence on interventions. None the less authors did a thorough job and should be commended.

Response: We would like to thank the reviewer for their encouraging comment and agree that engaging men in HIV care is of public health importance. The lack of an RCT may probably be due to limited work done among this population and we have recommended for more to be done in the conclusion.

Reviewer #2: The authors present a well written and rigorously conducted systematic review on the important topic of retention of heterosexual men in SSA. I have only a few minor comments:

 Response: We appreciate the reviewer for their comment.

Line 140 notes that articles among the excluded were "articles that did not measure the outcomes other than retention". I think this is confusing and the authors intend to exclude articles that measured outcomes other than retention.

Response: We would like to thank the reviewer for this observation. The sentence has been updated to clarify that articles that measured other outcomes other than retention.

Also the text shows the range of population of males in the examined studies. A description of the percentages of males would be of value in this section (around line 194).

Response: We thank the reviewer for this suggestion. The percentage of males has been included in the revised manuscript as a proportion of 32% of the total population. 

If males are more mobile, it may be that attrition or lost to follow up at a particular clinic or program may be followed by a resumption of care elsewhere and true retention deficits are overestimated. This is a challenge of much programmatic data because it is hard to understand true rates of attrition, is probably a shortcoming of included articles, and is probably worth addressing here. 

Response: We appreciate the reviewer for this observation and insight into this review. The suggestion has been included as one of the limitations of this review.

It may be interesting to note how SSA continuum outcomes compare to those in the US - that retention is not just a problem in SSA and some of the lessons on retention could be evaluated (with cultural caveats) for strategies to address gaps in the care cascade in resource rich settings as well. This is just a suggestion for consideration.

Response: We appreciate the reviewer for this observation and suggestion. We have included information on the USA as a comparison as part of the discussion section in the revised manuscript. 

Reviewer #3: The manuscript was presented in a very lucid and objective manner. There are sufficient terms for the study subject, and is properly guided with available literature which was well done The statistical analyses conducted represents the appropriate rigorous technical standards, well supported by the correct statistical techniques, which was described in adequately for clear understandable results.

As for the conclusions, this was also written appropriately based on available research, data and the aims and objective of the study.

The research is in line with the expected ethical standards necessary for the integrity of the study and publication.

Response: We wish to thank the reviewer for their valuable and encouraging comments on our review.

Reviewer #4: 

Comment 1: Review title and review objective are not inline to each other and need to be edited. Title: Strategies for retention of heterosexual men in HIV care in sub-Saharan Africa: A systematic review. Objective: This review summarizes attrition rates and interventions that reduce attrition of men in SSA cohorts.

Response: We wish to thank the reviewer for this comment. The review objective has been revised to; to summarize any studies on strategies to improve retention of heterosexual males (≥15years of age) in HIV care in SSA

Comment 2: Under abstract: Add number of articles found during search, whether PRIMA guideline following or not. In conclusion: 

Response: We appreciate the reviewer for this suggestion. The updated number of articles found during the search and that the PRISMA guidelines were followed has been added to the revised manuscript.

Comment 3: PICO is not well formulated. E.g.

• Line 118: I - intervention – interventions that aimed to improve retention among men living with HIV in SSA—is not focused to Community based strategies or Facility based strategies or both

Response: We appreciate the reviewer for their suggestion. However, the review focused on finding any strategy that would improve retention among men in HIV care and not limited to specific settings. 

• Line 120-121: says “O - outcome –Studies that documented retention proportion/rate, attrition rate/proportion, relative risk, hazard ratios, odds ratios or retention strategies”. According the title of review it is expected that the outcome of review is only retention strategies.

Response: We thank the reviewer for their comment. However, we listed statistical outcomes that are used measure whether a strategy/intervention works or not. The magnitude of the statistical measure would give an indication of how well it works.

Comment 4: The authors have done a search of studies published between January 2005 and April 2019. It is almost one year. It is better to update search at least up to Nov. 30 to get the current strategies and comprehensive strategies. 

Response: We appreciate the reviewer for this suggestion. The search was updated to include articles published up to 04 December 2020 as was also suggested by the Journal Editor.

Comment 5: Excluding published in languages other than English is not recommended in systematic review because it increase chance of publication bias. Report how many of the articles were excluded because of language restriction.

Response: We wish to thank the reviewer for this suggestion. Following the update made to our search, the limit for language was removed. The search returned only articles in the English Language. 

Comment 6: Although authors plan to use Cochrane risk of bias which comments for intervention studies, majority or all of the included studies are cohort study. Therefore, Newcastle Ottawa quality assessment for cohort study or STROBE checklist is an appropriate to assess risk of bias in this study. If the assume Cochrane risk of bias tool is appropriate they should present the graph the tool.

Response: We appreciate the reviewer for their comment and suggestion. We agree that all the studies included are cohort studies. We did not use the risk of bias tool (RoB), but rather the Cochrane Risk of Bias in Non-randomised Studies - of Interventions (ROBIN-I) which may include observational studies like: cohort studies, case control studies and others. Since our study had mainly cohort studies, we decided to use the ROBIN-I tool to assess risk of bias for this review.

We have included the graph below for our review in the revised manuscript.

 

 Study

 Decroo, 2017[15] Wringe, 2018[16] Dercoo, 2014[26] Fatti, 2012[27] Nabaggala, 2018[28] Rich, 2012[29] Tsondai, 2017[30] Kipp, 2012[31] Megereso, 2016[32] Akilimali, 2017[33] Siril, 2017[34] Shearer, 2016[35] Bilinski, 2017[36] Bock, 2019 [37]

Risk of bias domains Confounding - - - + - - - - - - - - - -

 Bias in selection of participants into the study - - x + - - + - - - - - - +

 Bias classification of intervention + - - - - - - - - - - - x +

 Bias due to deviations from intended interventions - - - - - - - - - + - - - +

 Bias due to missing data - - - - - - - x - x - + - -

 Bias in measurement of outcomes - - - - - - - - - - - + - -

 Bias in selection of the reported result + + + + + + - + + - + + + +

Key

+= Low risk of bias

_ = Moderate risk of bias

X= Serious risk of bias

Comment 7: My Big doubt: In searching strategies the authors include “Men or male or man or males OR "adult men" OR "Adult male" ”. Although this seems better to be specific to study population, it exactly excluded those study focus to both male and female and affect comprehensiveness of search.

Response: We appreciate the reviewer for their comment. The focus of this review was on males. However, during our search we found mainly studies that focused on both females and males. 

Comment 8: In table -2 a column with intervention need to be edited as it indicating the strategies of retention for each study. E.g. For Decroo, 2014, Mozambique [26], the strategies need to be edited as joint community ART groups rather than as impact of CAG. Some of the included strategies of retention is not clear e.g. 6monthly visits, Identification status and Distance to health center, 8Kms.

Response: We wish to thank the reviewer for their observation. The interventions have been edited and made clearer as suggested in the revised manuscript.

Comment 9: Line 242 says “What is interesting is no study was found that focused on men as a population?” According to this statement to the authors missed the review target population (male whose age is greater than or equal to15 years).

Response: We thank the reviewer for their observation. The line has been revised to what the target population for this review is, that is heterosexual males in care.

Comment 10: Line 253-257 talking about homosexual issue which is not related to this systematic review study (about heterosexual male). So need to be removed from study.

Response: We thank the reviewer for their suggestion. However, as part of the discussion, we were comparing retention rates among homosexual and heterosexual males so as to provide insight into the fact that men in general have low retention ion care. 

Comment 11: There is not table 1 in result section. 

Response: We appreciate the reviewer for this observation. The tables have been re-labeled from Table 1 (Studies that met the inclusion criteria), Table 2 (Risk of bias assessment)

Comment 12: Line 156: “SK and RN???” and line 182: “Titles title and…” need to be edited. 

Response: We thank the reviewer for their critical observation. The initials SK and RN have been removed, and the second word title has been deleted in the revised manuscript.

Comment 13: Add at least the keyword/text words and MeSH term used during searching for each concepts.

Response: We wish to refer the reviewer to the Supporting Information Table (S1 Table) that has details of the keywords used during the search. We used the common MeSH terms listed in publications related to retention in HIV care.

Comment 14: The authors should perform quality assessment of the original articles included in systematic review using appropriate quality assessment tools. 

Response: We thank the reviewer for their suggestion. The risk of bias assessment (quality assessment) was conducted for this review using the Cochrane ROBIN-I tool. 

Comment 15: Discussion section is not presented 

Response: We wish to refer the reviewer to the journal guidelines where results and discussion are presented as one. We have included the required section entitled Results and discussion in the updated manuscript.

Comment 16: The systematic review was not well concluded (add the reported intervention or strategies to retain male in HIV Care center).

Response: We thank the reviewer for their suggestion. We have since added a conclusion to the text.

Conclusion

This review suggests that no studies have explored heterosexual male centered interventions in HIV care. However, in included studies that explored retention in both males and females, there were high rates of attrition in men. The barriers and facilitators for retention of men in HIV care need to be explored in order to design male-centered interventions in SSA. There is also need to study the effectiveness of potentially effective strategies, preferably through randomized controlled trials and any interventions put in place should be evaluated

---

## [Decision Letter · Decision Letter 1]

20 Jan 2021

Strategies for retention of heterosexual men in HIV care in sub-Saharan Africa: A systematic review

PONE-D-20-19143R1

Dear Dr. Kusemererwa,

We’re pleased to inform you that your manuscript has been judged scientifically suitable for publication and will be formally accepted for publication once it meets all outstanding technical requirements.

Kind regards,

Claudia Marotta

Academic Editor

PLOS ONE

Additional Editor Comments (optional):

dear authors congratulations

Reviewers' comments:

Reviewer's Responses to Questions

**Comments to the Author**

1. If the authors have adequately addressed your comments raised in a previous round of review and you feel that this manuscript is now acceptable for publication, you may indicate that here to bypass the “Comments to the Author” section, enter your conflict of interest statement in the “Confidential to Editor” section, and submit your "Accept" recommendation.

Reviewer #1: All comments have been addressed

Reviewer #2: All comments have been addressed

2. Is the manuscript technically sound, and do the data support the conclusions?

Reviewer #1: Yes

Reviewer #2: Yes

3. Has the statistical analysis been performed appropriately and rigorously? 

Reviewer #1: N/A

Reviewer #2: Yes

4. Have the authors made all data underlying the findings in their manuscript fully available?

Reviewer #1: Yes

Reviewer #2: Yes

5. Is the manuscript presented in an intelligible fashion and written in standard English?

Reviewer #1: Yes

Reviewer #2: Yes

6. Review Comments to the Author

Reviewer #1: This is a much improved version of the manuscript and hope the paper and findings do lead to properly designed studies that can help improve engagement of men in HIV prevention and treatment. Congratulations on a nicely written review

Reviewer #2: This is a well-executed review. All my comments have been adequately addressed and I appreciate the author's attention to the issues raised.

7. PLOS authors have the option to publish the peer review history of their article (what does this mean?). If published, this will include your full peer review and any attached files.

Reviewer #1: **Yes: **Moses H. Bateganya

Reviewer #2: No

---

## [Editor Report · Acceptance letter]

22 Jan 2021

PONE-D-20-19143R1 

Strategies for retention of heterosexual men in HIV care in sub-Saharan Africa: A systematic review 

Dear Dr. Kusemererwa:

I'm pleased to inform you that your manuscript has been deemed suitable for publication in PLOS ONE. Congratulations! Your manuscript is now with our production department. 

Kind regards, 

on behalf of

Dr. Claudia Marotta 

Academic Editor

PLOS ONE